# MicroRNA Roles in Cell Reprogramming Mechanisms

**DOI:** 10.3390/cells11060940

**Published:** 2022-03-10

**Authors:** Emilia Pascale, Carmen Caiazza, Martina Paladino, Silvia Parisi, Fabiana Passaro, Massimiliano Caiazzo

**Affiliations:** 1Department of Molecular Medicine and Medical Biotechnology, University of Naples “Federico II”, Via Pansini 5, 80131 Naples, Italy; emilia.pascale@unina.it (E.P.); carmen.caiazza@unina.it (C.C.); martina.paladino@unina.it (M.P.); silvia.parisi@unina.it (S.P.); 2Department of Pharmaceutics, Utrecht Institute for Pharmaceutical Sciences (UIPS), Utrecht University, Universiteitsweg 99, 3584 CG Utrecht, The Netherlands

**Keywords:** non-coding RNAs, miRNAs, iPSCs, neuronal reprogramming, cardiac reprogramming

## Abstract

Cell reprogramming is a groundbreaking technology that, in few decades, generated a new paradigm in biomedical science. To date we can use cell reprogramming to potentially generate every cell type by converting somatic cells and suitably modulating the expression of key transcription factors. This approach can be used to convert skin fibroblasts into pluripotent stem cells as well as into a variety of differentiated and medically relevant cell types, including cardiomyocytes and neural cells. The molecular mechanisms underlying such striking cell phenotypes are still largely unknown, but in the last decade it has been proven that cell reprogramming approaches are significantly influenced by non-coding RNAs. Specifically, this review will focus on the role of microRNAs in the reprogramming processes that lead to the generation of pluripotent stem cells, neurons, and cardiomyocytes. As highlighted here, non-coding RNA-forced expression can be sufficient to support some cell reprogramming processes, and, therefore, we will also discuss how these molecular determinants could be used in the future for biomedical purposes.

## 1. Introduction

Cell reprogramming became an increasingly popular field after the discovery of induced pluripotent stem cells (iPSCs) [1]. This cell reprogramming milestone study from Takahashi and Yamanaka proved that the expression of key transcription factors (TFs), Oct4, Sox2, c-Myc, and Klf4, is sufficient to convert differentiated cells into induced pluripotent stem cells (iPSCs) [1]. This new experimental paradigm paved the way for using pluripotent stem cells in biomedical applications, including disease modeling, cell-based therapy, and tissue engineering [2,3,4].

Cell reprogramming approaches are not limited to iPSCs, indeed another technology named “direct reprogramming” has been developed in the last decades. Direct reprogramming, a transdifferentiation process, allows the direct conversion of somatic cells into other mature cell phenotypes without passing through a pluripotent state. Direct reprogramming can be achieved by overexpression of TFs, microRNAs (miRNAs), and/or treatment with small molecules. The first example of direct cell reprogramming mediated by TF-forced expression was the conversion of fibroblasts into muscle cells achieved by Dr. Weintraub and collaborators [5,6]. Since this groundbreaking discovery, researchers identified different reprogramming approaches that allow the generation of macrophages, hepatocytes, Sertoli cells, pancreatic beta cells, and other cell types that could be implemented for biomedical interests [7,8]. One of the most striking achievements of direct reprogramming is the generation of induced neurons (iNs) by imposing the overexpression of specific neurogenic TFs that can convert fibroblasts into functional neurons [9,10,11,12,13,14,15,16,17,18]. Another promising accomplishment in the cell reprogramming field is the generation of induced cardiomyocytes (iCMs), mediated by the overexpression of selected cardiogenic TFs [19,20]. Both these direct reprogramming approaches hold considerable potential for biomedical applications and have been perfected in the last ten years. The improvement of cell reprogramming methods is clearly linked to a deeper understanding of its molecular mechanisms. Indeed, a growing amount of literature has recently focused on the molecular determinants that are modulated by the TFs that can drive cell reprogramming processes. Among these, miRNAs have been increasingly investigated for their role in several types of cell reprogramming and especially in the generation of iPSCs, iNs, and iCMs. In this review we provide a brief overview of iPSC, iN, and iCM reprogramming processes, and we cover the main works that identified miRNAs involved in such reprogramming approaches, including a perspective on their potential use for biomedical applications.

## 2. iPSC Reprogramming Approach

In 2006, Takahashi and Yamanaka, demonstrated that iPSCs can be generated from terminally differentiated skin fibroblasts [1]. Afterwards, different mouse and human cell types, including keratinocytes, melanocytes, hepatocytes, astrocytes, neural stem cells, T cells, blood stem cells, and urine cells, have been reprogrammed into iPSCs [2].

Both mouse and human iPSCs showed many genetic and functional similarities with ESCs, such as patterns of gene expression and chromatin methylation, cell division rate, and the ability to form embryoid bodies, teratomas, and, in the case of mouse iPSCs, chimeras, thus demonstrating that iPSCs can contribute to all body tissues [21,22]. Moreover, significant research progress has been made to produce high quality iPSCs with greater efficiency [23]. However, the reprogramming process towards iPSCs causes a widespread epigenetic erasure of the original cells, thus resulting in a rejuvenation that could affect some applications for disease modeling [24]. Moreover, iPSCs were initially generated by integrating viruses bearing the risk of genomic damage and are, therefore, sub-optimal for cell-based therapies. To circumvent this problem, a variety of non-integrating methods have been employed, such as Sendai virus, recombinant proteins, synthetic messenger RNAs, small molecules, and plasmids [25,26,27]. Interestingly, several miRNAs and small molecules have been proven to enhance reprogramming efficiency or to be able to replace one or more of the essential reprogramming factors to obtain transgene-free iPSCs [26,27].

### miRNA in iPSC Reprogramming

The discovery of miRNAs as key modulators of ESC gene regulatory networks [28,29,30,31,32] also allowed the identification of several candidate miRNAs that could influence cell reprogramming towards iPSCs [33]. Dr. Blelloch’s group provided one of the first demonstrations that miRNA overexpression could enhance the efficiency of iPSC generation, as proven in the case the miR-290 cluster [33]. On the other hand, specific miRNA families have been proven to also negatively regulate the pluripotency network in ESCs [34]. In particular, the miRNA let-7 family has been clearly characterized for its capacity to inhibit ESC self-renewal [34]. Accordingly, the RNA binding protein LIN28 is able to downregulate let-7, therefore promoting the generation of iPSCs [35]. Indeed, in this study, Yu and colleagues demonstrated that the overexpression of LIN28, together with the TF NANOG, can substitute KLF4 and c-MYC in the Yamanaka reprogramming cocktail [35].

Eventually, several groups managed to prove that TFs can be completely replaced by defined ESC-specific miRNAs to reprogram human or mouse somatic cells to iPSCs [36,37]. This striking result was obtained by employing ESC-specific miRNAs identified by differential expression screenings. Specifically, to achieve miRNA-based cell reprogramming, it is sufficient to deliver the miR-302/miR-367 cluster [37] or the combination of hsa/mmu-miR-200c/hsa/mmu-miR-302s/hsa/mmu-miR-369s [36].

This is indeed in agreement with the previous evidence that the miR-302 family is directly induced by the TFs of the pluripotency network [31] and can facilitate the generation of iPSCs by inhibiting epithelial-mesenchymal transition (EMT), the cell cycle, epigenetic modifications, and the TGF-β pathways [38,39].

Mechanistic insights into the role of the miR-200 cluster during iPSC reprogramming showed that it directly binds to ZEB2 3′-UTR, stimulating mesenchymal-epithelial transition and accelerating the initiation phase of reprogramming [40]. An additional target for reprogramming inducing miRNAs is represented by p53, a well-known tumor-suppressor gene whose activation represents a checkpoint for the reprogramming process [41,42,43]. Among several miRNAs that are vital components of the p53 pathway, miR-138 directly targets the 3′-UTR of the p53 mRNA and significantly increases reprogramming efficiency [44].

In the last decade of research in this field, several other miRNAs have been identified for their ability to influence iPSC reprogramming. The overexpression of two members of the miR-106b-25 cluster and the miR-106a-363 clusters significantly enhance iPSC generation, while knockdown of the same miRNAs as well as another member of the same cluster, mmu-miR-25, decreases reprogramming efficiency [45]. Another parallel screening study identified several miRNAs, including the miR-130/miR-301/miR-721 family that were found to enhance the reprogramming of fibroblasts into iPSCs [46]. More recently, it has been proven that the miR-181 family is transiently induced during the initial phase of iPSC reprogramming and is subsequently silenced in iPSCs. The addition of miR-181 is able to reduce epigenetic barriers in the first phase of reprogramming, thus leading to an increase in the kinetics and efficiency of iPSC generation [47].

The main papers that highlighted the crucial role of miRNAs in iPSC reprogramming of mouse and human somatic cells are summarized in Table 1 and Figure 1.

## 3. Neural Direct Reprogramming Approach

Studies on CNS development and cellular reprogramming have provided the knowledge that modulation of TFs and/or miRNAs and small molecules are able to induce specific neuronal phenotypes [48]. In this context, the Dr. Wernig group demonstrated that dermal fibroblasts can directly be converted to functional iNs by means of ASCL1, BRN2, and MYT1L (and NEUROD1 for human cells) [9,10]. Subsequently, other TFs have been employed by others to generate specific neuronal subtypes [14,16,18,49], and miRNAs [11] as well small molecules [50,51] have been shown to promote iN generation.

Neural direct reprogramming, like other reprogramming methods, has been initially achieved using viral overexpression of TFs [9,10,11]. More recently, in order to increase the iN safety profile, several groups delivered the neurogenic TFs using plasmids [52], proteins [53], and non-integrating viruses [54].

Direct neuronal reprogramming has advantages compared to iPSC-derived neurons, including a fast route from fibroblast to the neuron of interest (~2–3 weeks) and maintenance of the epigenetic signatures of the donor cell. The latter point is of particular interest for the modeling of late-onset neurodegenerative disorders, such as Alzheimer’s disease (AD), Parkinson’s disease (PD), and amyotrophic lateral sclerosis (ALS) [55,56,57]. On the other hand, direct neural reprogramming also brings some limitations, such as a higher degree of variability in the converted cells due to the absence of clone selection as for iPSCs. Another downside of this reprogramming approach is linked to the relatively small number of studies focusing on its molecular mechanisms. To date, we know from transcriptomic and epigenomic studies that ASCL1 is the key TF with a pioneering activity that allows the remodeling of specific chromatin territories in the starting somatic cells [58,59,60]. Moreover, it has been shown that the generation of dopaminergic iNs (iDANs) with the ASCL1, NURR1, and LMX1A cocktail can be modulated by other molecular determinants, such as miRNAs and L1 mobile elements [61,62]. Intriguingly, chromatin remodeling has also been described in neural direct conversions based only on miRNA modulation [63,64,65]. Therefore, miRNAs are earning increasing attention in the process of direct neuronal reprogramming.

### 3.1. miRNAs in Direct Neuronal Reprogramming In Vitro

Several miRNAs have been identified as molecular determinants of neuronal phenotypes, including let-7, miR-9-5p/3p, miR-124, and miR-218 [66]. Indeed, the let-7 miRNA family is both an inhibitor of pluripotency and a promoter of the neural lineage [67]. miR-9-5p/3p and miR-124 interact with the gene regulatory network, inducing the specific expression of a neural differentiation program controlling BAF chromatin remodeling complexes during neural development [68]. This experimental evidence paved the way to employ has-miR-9-5p/3p and has-miR-124 in direct neural reprogramming. Indeed, a subsequent study from the same group proved that has-miR-9-5p/3p and has-miR-124 promote the direct conversion of human adult fibroblasts towards neurons, a process greatly enhanced by co-expressing TFs, NeuroD2, ASCL1, and MYT1L, that increase the rate of conversion and maturation into MAP2- positive glutamatergic neurons [11]. Following studies from Dr. Yoo’s group who reported that co-expression of has-miR-9-5p/3p and has-miR-124 with TFs enriched in the developing striatum, such as CTIP2, DLX1, DLX2, MYT1L, can guide the conversion of fibroblasts into striatal GABAergic medium spiny neurons [13].

More recent evidence showed that miR-9-5p/3p-miR-124 are able to induce chromatin access to a wide number of loci that drive neuronal subtype differentiation [63]. Consistently, over-expression of has-miR-124 and has-miR-9-5p/3p, together with TFs that drive towards motor neuron differentiation (ISL1 and LHX3), promotes a reprogramming into a motor neuron phenotype [63].

Interestingly, miR-9-5p/3p-miR-124 are indirectly negatively regulated by polypyrimidine tract-binding proteins 1 and 2 (PTBP1-2) in neural cells [69,70]. Therefore, the repression of PTB proteins was found to be sufficient to convert fibroblasts into functional neurons [69,70].

### 3.2. miRNAs in Direct Neuronal Reprogramming In Vivo

More recently, other miRNAs have been shown to improve reprogramming processes both in vitro and in vivo. This evidence has been clearly shown in the case of direct neural reprogramming into dopamine neurons. De Gregorio and colleagues showed that miR-34b-5p/miR34c-5p, when overexpressed together with ASCL1 and NURR1, can increase the efficiency of reprogramming into iDANs [61]. On the other hand, a seminal paper in the field of in vivo neuronal direct reprogramming showed that conversion into iDANs is enhanced when miR-218 is delivered together with ASCL1, NEUROD1, and LMX1A [71].

The main papers that highlighted the crucial role of miRNAs in the direct neuronal reprogramming of mouse and human somatic cells are summarized in Table 2 and Figure 1.

### 3.3. Challenges and Opportunities of miRNA-Mediated Neuronal Reprogramming as a Therapeutic Strategy to Treat Neurodegenerative Diseases

miRNAs fulfill a well-known role in the normal development of the central nervous system and in targeting gene expression, acting as key regulators in several neuroprotective mechanisms. The recent developments of non-coding RNA biology have quickly projected this field towards biomedical applications for neurodegenerative diseases. Alterations in the miRNA expression profile can be associated with the stage of neurodegenerative diseases and could be used as diagnostic biomarkers of brain function [72,73]. On the other hand, since the groundbreaking work by Rivetti di Val Cervo and colleagues [71], in vivo direct neural reprogramming has been considered a promising addition to the gene therapy approaches that could be used to treat neurodegenerative diseases [74,75]. In this context, miRNA modulation has been explored to treat animal models of PD by silencing PTB proteins, therefore inducing the neurogenic miRNAs miR-124 and miR-9-5p/3p. Two recent studies provided the evidence that such an approach could be used to reprogram brain-resident astrocytes into iDANs, either by delivering a PTBP1 antisense oligonucleotide or by depleting PTBP1 mRNA with CRISPR-CasRx [76,77]. Strikingly, both works reported that the brain in situ conversion into iDANs is able to rescue the motor behavior of a PD animal model [76,77]. Anyways, it has to be mentioned that these extraordinary results based on a possible in situ astrocyte-to-neuron conversion have been recently doubted after using a more stringent astrocyte lineage tracing [78]. Therefore, although the use of miRNA modulation for in vivo therapeutic approaches represents a logical option, clearly more work remains necessary to develop it into a reliable treatment for PD or other neurodegenerative diseases.

## 4. Direct Cardiac Reprogramming Approach

Over the last decade, significant progress has been made in the development of novel therapeutic approaches based on the direct cardiac reprogramming of a patient’s somatic cells. These efforts eventually led to the generation of induced cardiomyocytes (iCMs) that could be used to repair a damaged fibrotic myocardium [79,80] with new contractile cells without passing through a pluripotent state [81,82]. In this contest, the ideal strategy would be to reprogram cardiac fibroblasts (CFs), which account for up to 11% of the heart tissue, into functional iCMs [83,84].

To date, direct cardiac reprogramming has been achieved for several human and mouse cell types by the forced expression of TFs and/or non-coding RNAs or through the delivery of small molecules modulating crucial pathways [81,82] (Table 1).

Most reprogramming cocktail combinations include lineage-specific pioneer TFs, which bind and open condensed chromatin loci to recruit other canonical TFs [85]. GATA4 is the pioneer for both mouse and human cardiac reprogramming [19,86,87,88] and cooperates with other TFs to synergistically activate in fibroblasts a cardiomyocyte gene program [89]. The combination of GATA4, MEF2C, and TBX5, the so-called GMT reprogramming cocktail, was the first and more effective in inducing mouse direct cardiac reprogramming [85], followed by several attempts based on adding other TFs to the GMT core, such as HAND2 (GHMT cocktail) [88], NKX2.5 [90], or a combination with serine/threonine kinase AKT1 [91].

More recently, direct cardiac reprogramming protocols have been refined by integrating GMT overexpression with the modulation of polycomb repressive chromatin remodeling complexes 1 (PRC1) [92,93] or 2 (PRC2) [94,95].

Several approaches in direct cardiac reprogramming adopted the addition of small molecules to the GHMT cocktail to inhibit pro-fibrotic signaling and enhance cardiac fate, such as inhibitors of the TGFβ [96,97] and Notch pathways [98]. Anti-inflammation may represent another potential critical target for lineage conversions, especially from the perspective of clinical translation [80,99], as the inhibition of pro-inflammatory cytokines ameliorates direct cardiac reprogramming efficiency [93,100].

Despite the excitement for the potential of direct cardiac reprogramming, the approach still presents many limitations, such as low yield and maturation of iCM and the use of unsafe viral vectors. A promising hope to improve the generation of iCMs comes from the identification of several miRNAs that regulate cardiac regeneration and are considered to be potential therapeutic targets [101].

### 4.1. miRNA-Mediated Direct Cardiac Reprogramming In Vitro

MiRNAs have emerged as functional regulatory molecules in direct cardiac reprogramming, managing processes such as cell cycle progression, DNA methylation, and cell differentiation. As such, they have been used as an alternative to TFs overexpression. In 2012, Jayawardena et al., by screening the potential miRNAs involved in CM development and differentiation, found that a combination of four miRNAs, called miRcombo (mmu-miR-1, mmu-miR-133a, mmu-miR-208a, and mmu-miR-499), was able to promote in vitro direct cardiac reprogramming of mouse fibroblasts into iCMs [102]. The addition of a Janus kinase 1 (JAK1) inhibitor to the miRcombo further increased the reprogramming efficiency [102]. The bicistronic miRs, mmu-miR-1 and mmu-miR-133a, regulated by Mef2, were already known to play common roles in regulating CM proliferation and ventricular organization in developing cardiac and skeletal muscle tissues, whereas mmu-miR-208a and mmu-miR-499, encoded by myosin heavy chain genes, were mostly associated with the regulation of CM contraction, hypertrophy, and cardiac stress response pathways [103].

Following this first attempt, several groups tried to exploit the benefits of cardiac miRNAs in promoting cell fate conversion. Dal-Pra and colleagues found that the overexpression of miRcombo induces the upregulation of lysine demethylases 6A (KDM6A) in neonatal CFs by downregulating EZH2 gene expression. This, in turn, upregulates the expression of endogenous GHMT reprogramming factors [104].

Mouse fibroblasts have been directly reprogrammed into cardiomyocytes, endothelial cells, or smooth muscle cells using mmu-miR-208b-3p, ascorbic acid, and bone morphogenetic protein 4 (BMP4) [105]. Once implanted into infarcted mouse hearts, reprogrammed cells were able to reduce tissue injury, improving cardiac function, since cardiomyocytes, which initially displayed immature characteristics, became mature over time and formed gap junctions with host cardiomyocytes [105].

Recently, it has been demonstrated that miRcombo alone can effectively reprogram human fibroblasts into iCM as well [106], although with less efficiency compared to mouse direct cardiac reprogramming.

Combining miRNAs with TFs by mixing viral/non-viral approaches also increased reprogramming efficiency in both MEFs and human CFs. In MEFs, the addition of mmu-miR-133a to GMT increased the generation of spontaneously contracting cells expressing the sarcomere protein α-actinin [107] through the repression of Snai1, a master regulator of the epithelial-to-mesenchymal transition [107]. Similarly, the addition of mmu-miR-1 and mmu-miR-133a to GHMT dramatically increased the percentage of spontaneously beating cells with respect to GHMT alone [97]. In addition, the parallel inhibition of TGF-β or ROCK signaling was found to be effective to further increase beating iCMs upon GHMT + mmu-miR-1 + mmu-miR-133a transduction [108].

Human CFs have been reprogrammed by the addition of mmu-miR-133a to a GMTMM (GMT + MESP1 + MYOCD) cocktail [107]. Others demonstrated that human iCM could also be generated by adding mmu-miR-1 and mmu-miR-133a to the GHMT cocktail [109] or to a GMT-MYOCD-NKX2.5 cocktail [110]. This latter study also showed a dramatic increase in the percentage of spontaneously contracting iCM, with a significant upregulation of cardiac gene signatures following the addition of JAK1 and GSK3β inhibitors [110].

GMT, in combination with miR-590, induced the expression of the cardiomyocyte marker cardiac troponin T (cTnT) in ~5% of human or porcine CFs [111], and this efficiency was further increased, combining the dual inhibition of HDACs and WNT with retinoic acid administration alongside GMT plus hsa-miR-590 transduction [112].

The use of Sendai virus to actively deliver GMTMM + mmu-miR-133a to human CFs further demonstrated the potential efficacy of miRNA and TF combinations in inducing direct cardiac reprogramming [113], although it has been recently reported that the administration of a polycistronic vector ensuring miRcombo delivery at a stoichiometric ratio can be more effective compared to other viral methods in inducing direct cardiac reprogramming [114].

### 4.2. miRNA-Mediated Direct Cardiac Reprogramming In Vivo

Jayawardena and colleagues were the first to prove that miRNAs can directly reprogram cardiac fibroblasts into cardiomyocytes in vivo [102].

miRcombo delivery using a lentiviral vector also induced direct cardiac reprogramming in vivo in ischemic mouse hearts [115]. A lineage tracing analysis by Fsp1Cre-traced fibroblasts demonstrated that iCM were most likely of fibroblastic origin. More interestingly, improved cardiac functions could be associated with reprogrammed cells up to 3 months after myocardial infarction (MI) [115].

The use of Sendai virus to actively deliver in vivo GMTMM + mmu-miR-133a to human CFs further demonstrated the potential efficacy of miRNA and TF combinations in inducing in vivo direct cardiac reprogramming [113], although it has been recently reported that the administration of a polycistronic vector ensuring miRcombo delivery at a stoichiometric ratio can be more effective compared to other viral methods in inducing direct cardiac reprogramming [114].

Overall, this evidence indicates that miRNA-mediated direct cardiac reprogramming represents a fascinating and promising prospect for regenerating the distressed heart. However, there are still many challenges to overcome that limit its clinical application.

The main papers that highlighted the crucial role of miRNAs in the direct cardiac reprogramming of mouse and human somatic cells are summarized in Table 3 and Figure 1.

### 4.3. Challenges and Opportunities of miRNA-Mediated Cardiac Reprogramming as a Therapeutic Strategy to Treat Heart Failure Patients

The clinical application of miRNA-mediated direct cardiac reprogramming strategies in humans still faces the limit of a lower efficiency compared to mice, thus requiring the experimentation of additional factors. Currently, there are two general strategies to introduce miRNAs in the injured heart. One is the delivery of miRNAs via viral-vector-mediated overexpression. Retroviral and lentiviral vectors, frequently used to introduce reprogramming factors, can potentially be associated with a risk of tumorigenesis due to host cell genome damage, in particular when introducing multiple genes [116], and the development of immune reactions in human patients [117]. Adenoviral vectors have already been used in hundreds of clinical trials with no evidence of tumor formation in long-term follow up [118]. Nonetheless, they present a certain grade of heterogeneous tropism, which might compromise the cell target specificity [119]. Sendai virus, derived from an enveloped virus with a non-segmented negative-strand RNA genome, seems to be safer and not pathogenic to humans, as they do not integrate into the host genome and only replicate in the cytoplasm [113]. Nevertheless, Sendai virus, like other viral vectors, cannot allow the timing or the temporal release of reprogramming miRNAs, which would be of great importance.

To overcome this challenge, novel nanotechnology-based delivery methods might represent a solution, allowing both the delivery of miRNA mimics and precise cardiac targeting [80,84]. Nanocarriers would enable the delivery of reprogramming cargoes to the infarcted area to reduce cellular and systemic toxicity. A very recent attempt demonstrated that an miRcombo cargo loaded in polyethyleneimine-coated nitrogen-enriched carbon dots yields an efficient direct cardiac reprogramming of mouse CFs to iCMs, with functional recovery of infarcted hearts and a negligible toxicity [120]. Another interesting approach based on a non-viral biomimetic system used silicon nanoparticles coated with FH-peptide-modified neutrophil-mimicking membranes loaded with miRcombo. This system exploits the natural inflammation-homing ability of neutrophil membrane proteins, in addition to the high affinity of the FH peptide to the tenascin-C (TN-C) produced by CFs, to target and deliver the miRcargo directly into CFs in the injured heart, leading to efficient reprogramming and improved cardiac function [121].

Currently, none of the miRNA-based direct cardiac reprogramming treatments are in clinical trials. This is also due to the lack of studies on the long-term consequences of such novel cardiac regeneration strategies in large animal pre-clinical models of dilated cardiomyopathy and other chronic heart diseases, which require regenerative therapies.

Therefore, a required step toward clinical translation will be the development of novel miR-mediated therapeutic tools to bring them from the bench to the market.

## 5. Conclusions

In the last 10 years, a large variety of miRNA-based reprogramming approaches have been established in order to generate iPSCs, iNs, and iCMs. These alternative reprogramming routes could represent a very promising solution to translate cell reprogramming into therapeutic solutions. Notwithstanding the issues due to low reprogramming rates and some recent doubts about the efficiency of the in vivo neuronal reprogramming results, the therapeutic implementation of miRNAs still represents a potential step forward for future clinical applications. Indeed, the implementation of miRNAs into reprogramming strategies would bring technical delivery advantages.

Several systems are available to deliver miRNAs in vivo, including liposomes, polymers, expression plasmids, and viruses. The most efficient results for in vivo reprogramming have been achieved with integrating viruses (lentiviruses and retroviruses) but they have limited chance to be considered for human applications considering the risk of genomic damage. Modern delivery systems are highly desired for in vivo direct treatments, and liposomal and cationic polymeric delivery systems are promising solutions for both miRNA expression plasmids and mature miRNAs [122]. Delivered miRNAs can be easily combined within a DNA-based vector, whereas the simultaneous delivery of multiple TFs in the same cell can be challenging. On the other hand, mature miRNAs can be chemically modified and easily delivered using synthetic delivery systems that exhibit low toxicity [123,124]. To date, clinical trials for cancer applications have been conducted using miRNA mimics with liposomal formulations as in the case of mir-34a (MRX34, miRNATherapeutics) [122].

It has to be mentioned that miRNA therapeutics only recently entered into clinical trials [125] for cancer (MRX34) and fibrosis (Remlarsen) applications. If successful, these pioneering studies will possibly pave the way for other miRNA-based therapies, including reprogramming approaches that could move to a clinical landscape in the next decade.

## Figures and Tables

**Figure 1 cells-11-00940-f001:**
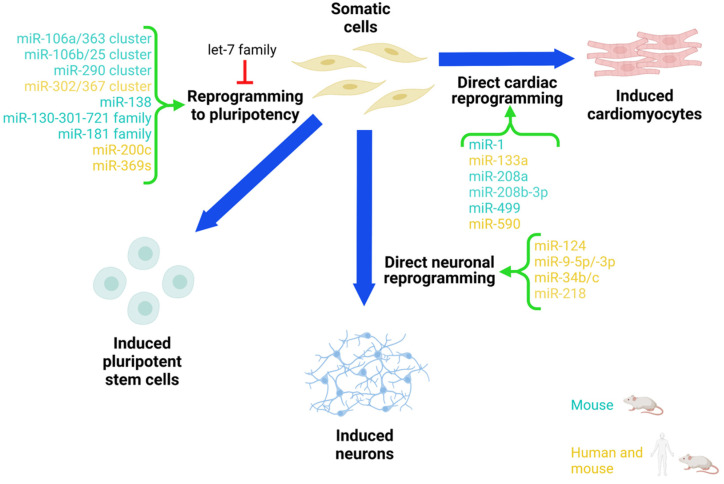
miRNAs involved in iPSC reprogramming and direct reprogramming to neurons and cardiomyocytes. The image shows miRNAs with a promoter role (in green) and others with an inhibitory role (in red) involved in the process of converting somatic cells into iPSCs and those miRNAs (in green) that promote the direct conversion of somatic cells into cardiomyocytes and neurons. These miRNAs can be used to improve the reprogramming efficiency.

**Table 1 cells-11-00940-t001:** Summary of the main miRNAs discussed in the text and their functions in iPSC reprogramming.

miRNAs	Functions in iPSC Reprogramming	Direct Targets	References
miR-290 cluster (mmu-miR-291a-3p, mmu-miR-294, and mmu-miR-295)	Significantly increases the number of mouse iPSC colonies	MEK pathway -directly repressing Akt1	[33]
miR-let7 family	Its inhibition via LIN28 facilitates induced pluripotency.	c-Myc, Lin28b, and Hmga2	[35]
miR-302/367 cluster	Induces pluripotency of mouse and human fibroblasts without exogenous expression of other transcription factors	TGFβ receptor 2 NR2F2	[37]
hsa/mmu-miR-200c, hsa/mmu -miR-302s, and hsa/mmu miR-369s	Used to induce pluripotency of mouse and human somatic cells without integration of any viral-based vectors	Aof1 and Zeb1/2	[36]
miR-130/ 301/721 family	Enhances the efficiency of iPSC generation by repressing the homeobox transcription factor Meox2.	Meox2	[46]
mmu-miR-138	Targets the 3’-UTR of the p53 mRNA and significantly increases reprogramming efficiency	p53 signaling pathway	[44]
miR-106b-25/ miR-106a-363 clusters	Overexpression of members of these two clusters significantly enhances iPSC generation.	Tgfbr2 and p21	[45]
miR-181 family	Is transiently induced during the initial phase of iPSC reprogramming	Cpsf6, Nr2c2, Marcks, Bptf, Igf2bp2, Nol8, Bclaf1, and Lin7c	[47]

**Table 2 cells-11-00940-t002:** Summary of the main miRNAs discussed in the text with their known functions in direct neuronal reprogramming.

miRNAs	Functions in Neuronal Cell Reprogramming	Direct Targets	References
hsa-miR-9-5p/-3p hsa-miR-124	Neuronal fate induction	PTPB1, REST, CoREST, SCP1, and BAF53a	[11]
hsa-miR-124/hsa-miR-9-5p/3p + CTIP2/DLX1/DLX2/MYT1L	Promotes the differentiation from human adult fibroblasts to GABA medium spiny neurons	PTPB1, REST, CoREST, SCP1, and BAF53a	[13]
hsa-miR-124/hsa-miR-9-5p/3p + ISL1/LHX3	Promotes the differentiation from human adult fibroblasts to motor neurons	PTPB1, REST, CoREST, SCP1, and BAF53a	[63]
miR-218 + ASCL1/NEUROD1/ LMX1A	Promotes in vivo astrocyte-to-neuron conversion	Onecut2	[71]
miR-34b-5p/miR-34c-5p + ASCL1/NURR1	Increases the generation of iDANs	Wnt1	[61]

**Table 3 cells-11-00940-t003:** Summary of the main miRNAs discussed in the text with their functions in direct cardiac reprogramming.

miRNAs	Functions in Cardiac Cell Reprogramming	Direct Targets	References
mmu-miR-1/mmu-miR-133a/mmu-miR-208a/mmu-miR-499	This combination (miRcombo) induces transdifferentiation to iCMs.	Twf1, Col16a1, and Ezh2	[102]
mmu-miR-208b-3p + ascorbic acid/BMP4	This combination (MAB) induces transdifferentiation to iCMs.	Gata4	[105]
mmu-miR-133a + GMT/GMTMM cocktails	Increases the efficiency of iCM generation	Snai1	[107]
mmu-miR-1/mmu-mmu-miR-133a + GHMT cocktail	Increases the efficiency of iCM generation	Twf1 and Snai1	[109]
hsa-miR-590 + GMT cocktail	Increases the maturation of iCMs	Sp1	[111]

## Data Availability

Not Applicable.

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
