# Peer review of "MicroRNA Roles in Cell Reprogramming Mechanisms"

_cells, 2022, doi:10.3390/cells11060940_

Round 1

Reviewer 1 Report

This is an interesting review on the use of miRNAs for the direct reprogramming of cells for biomedical interests. Although interesting, it is rather flimsy, being no more than a catalogue of miRNAs and their actions with poor mechanistic detail. Nevertheless, I think that this could be improved by following the next advices

1.- In general there’s no distinction of the reprogramming experiments made “in vitro” with cells or “in vivo” in animal models. Discussing these in different sections would make the review more interesting. Furthermore, the review would greatly benefit from including a section on a comparative of methods for transfecting miRNAs (liposomes, nanocarriers, expression plasmids or viruses, etc), as well as a discussion on the off-target effects of that manipulation. In this sense, I would recommend to re-write section 4.2 with these objectives in mind.

2-Species identifiers (hsa, mmu...) as well as the appropriate suffixes (a/b, 3p/5p) should be added to miRNA names for their proper identification.

3-Being that miRNAs mostly interact with 3’UTRs and that mRNAs are able to control the length of these 3’UTRs to regulate miRNA accession to their binding sites I would recommend to add a section on this very interesting topic covering reprogramming miRNAs.

Lastly, there are some mistakes in the text that should be corrected

-Belloch for Blelloch (line 80)

-3’UTR in Table 1 and line 105, but 3’ UTR (with space) in line 100

-Line 242, add space between “pathways” and the reference

-Reference numbers in tables are not correlative with regard of the rest of the text. The first reference in table 1 is numbered 33 when the last one in the corresponding text is number 47, reference numbers in tables 2 and 3 seem to be random.

Author Response

We are grateful for the reviewer comments that we addressed as follows:

1) We added new considerations on miRNA clinical applications that are now included in the conclusions.

2) We added the miRNA species in the text and in the tables.

3) We added in the tables a section that indicates the direct 3' UTR targets of the mentioned miRNAs

4) We made typing corrections in the text as indicated.

Reviewer 2 Report

The review is nicely and appropriately focused on the role of microRNAs in a specific topic: the cell reprogramming. This would perhaps restrict the audience of this manuscript; nevertheless, narrowing the focus is a wise and meaningful strategy in the mare magnum of experimental papers and reviews that have been published on the microRNA world.

The focus of this review is made clear in the title as well as in the Introduction, where the different approaches used in cell reprogramming are fully and orderly described and referenced.

Then, Authors have chosen to describe four models of cell reprogramming: i) The reprogramming of induced pluripotent stem cells, ii) The direct reprogramming of cells towards neurons and iii) The direct reprogramming toward cardiac cells. Each description is complete and gives significant information to describe the different approaches.

Three Tables are listing the necessary information of miRNAs whose role has been discussed in each preceding paragraph.

The final Figure 1 is schematically but clearly summarizing the different miRNAs that have been involved in the different reprogramming approaches.

The list of reference is very complete and exhaustive for the present knowledge on the role of miRNAs in the cell reprogramming approaches to which this review is focussed.

In summary, I think that the subject of this review is appropriately focused, the content is nicely organised in ordered paragraphs describing different approaches of cell reprogramming, followed by sub-paragraphs where the experimental data about the role of specific miRNAs that have been involved in different models of each approach. The text is easy to follow and very understandable. Bibliography is fully exhaustive.

Minor remarks

I think it would be valuable for the Reader if Author will introduce Tables with the basic information (miRNA code, function, reference) about other miRNAs that have been involved in the different reprogramming models chosen for this review, albeit not described (if there are some). Tables now list miRNAs that have been already discussed in the text and referenced in the bibliography. More complete Tables will give a comprehensive view of miRNAs involved and hints for the reader to those miRNAs that are not discussed in the text. As long as this does not make the Tables and the list of references too long.

Author Response

We are grateful to the reviewer for the positive comments.

Regarding reviewer raised point we did not find miRNAs that have been shown to have a major role in other reprogramming models.  On the other side we improved the tables indicating what is the direct molecular targeting role of the listed miRNAs.

Round 2

Reviewer 1 Report

Authors must correct a few typos regarding miRNA nomenclature in lines 160, 161, 165, 166, 171 and Table 2. They wrote has-miR.... instead of hsa-miR...